# Pre-Operative Malnutrition in Patients with Ovarian Cancer: What Are the Clinical Implications? Results of a Prospective Study

**DOI:** 10.3390/cancers16030622

**Published:** 2024-01-31

**Authors:** Sara Nasser, Esra Bilir, Xezal Derin, Rolf Richter, Jacek P. Grabowski, Paulina Ali, Hagen Kulbe, Radoslav Chekerov, Elena Braicu, Jalid Sehouli

**Affiliations:** 1Department of Gynecology with Center for Oncological Surgery, Charite Comprehensive Cancer Center, 13353 Berlin, Germanyelena.braicu@charite.de (E.B.);; 2Department of Global Health, Koç University Graduate School of Health Sciences, İstanbul 34450, Turkey; esragbilir@gmail.com

**Keywords:** malnutrition, ovarian cancer, overall survival, nutritional risk screening score, nutrition, phase angle, progression-free survival

## Abstract

**Simple Summary:**

In the literature, between 30% and 80% of all patients with cancer are reported to be malnourished or cachectic. Our objectives included identifying the risk factors for malnutrition in patients with ovarian cancer, determining the diagnostic relevance of the commonest methods used to assess nutritional status in these patients, and evaluating the predictive and prognostic values of malnutrition in patients with primary and relapsed ovarian cancer. We found malnutrition as an independent predictor of incomplete cytoreduction and an independent prognostic factor for poor overall survival. Preoperative nutritional assessment is an effective tool in the identification of high-risk groups of ovarian cancer characterized by poor clinical outcome.

**Abstract:**

Background: Malnutrition was associated with worse survival outcomes, impaired quality of life, and deteriorated performance status across various cancer types. We aimed to identify risk factors for malnutrition in patients with epithelial ovarian cancer (EOC) and impact on survival. Methods: In our prospective observational monocentric study, we included the patients with primary and recurrent EOC, tubal or peritoneal cancer conducted. We assessed serum laboratory parameters, body mass index, nutritional risk index, nutritional risk screening score (NRS-2002), and bio-electrical impedance analysis. Results: We recruited a total of 152 patients. Patients > 65 years-old, with ascites of >500 mL, or with platinum-resistant EOC showed statistically significant increased risk of malnutrition when evaluated using NRS-2002 (*p*-values= 0.014, 0.001, and 0.007, respectively). NRS-2002 < 3 was an independent predictive factor for complete tumor resectability (*p* = 0.009). The patients with NRS-2002 ≥ 3 had a median overall survival (OS) of seven months (95% CI = 0–24 months), as compared to the patients with NRS-2002 < 3, where median OS was forty-six months (*p* = 0.001). A phase angle (PhAα) ≤ 4.5 was the strongest predictor of OS. Conclusions: In our study, we found malnutrition to be an independent predictor of incomplete cytoreduction and independent prognostic factor for poor OS. Preoperative nutritional assessment is an effective tool in the identification of high-risk EOC groups characterized by poor clinical outcome.

## 1. Introduction

Malnutrition has been associated with worse survival outcome, impaired quality of life, and deteriorated performance status in patients across various cancer types [1,2,3]. The literature shows that approximately half of hospitalized patients are malnourished [4]. 

Although modern interventions are continuously being developed within cancer therapies, advances in assessing, preventing, and treating malnutrition still remain limited. Approximately 30–80% of all patients with cancer are malnourished or cachectic [4,5,6,7]. This is influenced by tumor type, location, stage, and current therapy [6,7,8]. Malnutrition can even be a direct cause of death in patients with advanced cancer stages [9,10]. It is only in recent years that studies have focused more specifically on evaluating malnutrition, sarcopenia, and cachexia in patients with gynecological cancers [7,9]. It has been shown that malnutrition has a higher prevalence in patients with gynecological cancers [9]. Accumulating data also reveal that malnutrition has a significant impact on survival outcomes, hospital stay, and postoperative complications in patients with gynecological cancers [9,10,11,12]. The reasons for malnutrition in this patient group are likely to be multifactorial, including metabolic effects of the disease process itself as well as reduced oral intake, socio-economic factors, age, and polypharmacy [9,11,12]. 

Although more data are emerging in patients with gynecological cancers, studies evaluating malnutrition specifically in ovarian cancer patients and its impact as a predictive and prognostic tool are still sparse.

Studies have shown that patients with ovarian cancer are at higher risk of malnutrition compared to any other gynecological cancers [5,7,9,13,14]. Protein-energy malnutrition and cachexia are diagnosed in up to 81.4% of all the patients with ovarian cancer [13,14,15,16,17,18]. Moreover, patients with ovarian cancer were found to be 19 times more likely to present with malnutrition compared to the patients with benign conditions [14]. At the time of diagnosis, 66.7% of patients with epithelial ovarian cancer (EOC) have been found to be malnourished [14] and showed higher complication rates and longer hospital stays [19]. Additionally, malnutrition has been associated with shorter overall survival in patients with EOC [15,20,21]. Consequently, these high-risk patients require a complex combination of nutritional assessment strategies and individualized treatment plans.

Currently, the validated tools for nutritional screening include Nutritional Risk Screening 2002 (NRS-2002), the Malnutrition Universal Screening Tool (MUST), Malnutrition Screening Tool (MST), and Mini Nutritional Assessment (MNA), in the literature [11,17,18,22,23]. Moreover, the Nutritional Risk Index (NRI) has been developed to assess malnutrition in surgical patients [17,18,22,23].

In our prospective observational monocentric study, we aim to identify risk factors for malnutrition in patients with EOC, determine the diagnostic relevance of the commonest methods used to assess nutritional status in these patients, and evaluate the predictive and prognostic values of malnutrition in patients with primary and relapsed EOC.

## 2. Materials and Methods

### 2.1. Study Design

This is a prospective observational single-center study of the patients with primary and recurrent EOC, tubal or peritoneal cancer conducted at a tertiary care comprehensive cancer center. Approval from the Ethics Committee was obtained under the following reference number: EA2/142/07. We recruited patients between February 2007 and October 2008. Inclusion and registration in the study occurred upon admission. All the data were collected until discharge with a follow-up period of up to three years after diagnosis or until death. All patients over 18 years of age with histologically confirmed EOC, tubal or peritoneal cancer in the primary or relapse setting, who were admitted for planned primary or secondary/tertiary cytoreductive surgery were included in the study. We obtained written consent from all the patients for their participation and inclusion in the tumor bank ovarian cancer (TOC) database. We excluded patients with non-epithelial ovarian tumors, with pre-existing pacemakers or implanted defibrillators (contraindications for bio-electrical impedance analysis (BIA) measurement), as well as pregnant or breast-feeding patients from our study.

### 2.2. Assessment of Nutritional Status

The nutritional parameters and malnutrition screening tools used in our study have all been previously validated in several other clinical studies [17,18,22,23]. We assessed patients’ nutritional status preoperatively on admission to classify patients as “malnourished” or “non-malnourished”. Our study focused on the objective methods for nutritional-status assessment. Venipuncture was always performed on the first day of admission, prior to any intravenous infusions or supplements. Patient baseline characteristics were collected, including past medical history and general clinical examination.

We determined the following nutritional-status indicators for each patient included in the study:(1)Serum laboratory parameters: hemoglobin (g/dL), lymphocytes(/nl), albumin(g/dL), pre-albumin (mg/L), transferrin (mg/dL), and C-reactive protein (CRP) (mg/dL).(2)Body mass index (BMI) and nutritional risk index (NRI) calculations were based on the following formulas: BMI = weight (kg)/(height (m))^2^ and NRI = (1.489 × Serum Albumin g/L) + 41.7 × (current weight/usual weight) [12]. During the recruitment phase of the study, our research team conducted measurements of both weight and height for all study participants. We classified the BMI values into four categories, as follows: <18.5 kg/m^2^ underweight, 18.5–24.9 kg/m^2^ normal weight, ≥25.0 kg/m^2^ overweight, and ≥30.0 kg/m^2^ obesity.(3)The Nutritional Risk Screening Score (NRS-2002), a validated score, was determined in each patient, to classify the risk for malnutrition [24]. We classified the patients with a score of ≥3 as high-risk for malnutrition.(4)BIA is a relatively simple, inexpensive and non-invasive technique to measure body composition [25]. Each patient underwent BIA to measure body composition.

We used the following equipment to perform the analysis: B.I.A 2000-M (Series No. 0706) measurement apparatus from Darmstadt GmbH, BIA Phasertabs Ag/AgCl electrodes from MEDI CAL Healthcare GmbH Karlsruhe, and the software Nutri Plus Version 5.1, Data Input GmbH Darmstadt. We followed the standard company instructions to obtain all the measurements. During the measurements, each patient was asked to lie supine at approximately 45 degrees. The electrodes were applied to the right hand and foot and then connected to the measurement device via color-coded wires, in accordance with the instruction manual. All measurements were made by the same investigator to eliminate the risk of inter-observer variability. Two main BIA-parameters were recorded: extra cellular mass (ECM)/body cell mass (BCM) index and the phase angle (PhAα).

In a healthy well-nourished adult, the BCM is always greater than ECM with an ECM/BCM index <1. BCM loss is mainly due to loss of muscle mass, which is secondary to increased catabolism, while a rise in ECM can be due to the third-space fluid losses from edema, renal or cardiac failure. Changes in the ECM/BCM index can occur before weight loss (WL) is detected and is a sensitive index of malnutrition [24].

The PhAα is the relation of the two impedance components at 50 KHz, reactance (X_c_) and resistance (R), expressed as PhAα = (X_c_ × 180°)/(R × π). In female patients a value of ≥5 degrees is considered adequate.

The cut-off values for each nutritional status indicator, at which patients were classified as “malnourished”, were defined as follows, based on corresponding receiver operating characteristic (ROC)-curve analyses: NRS-2002 ≥ 3, NRI < 100, albumin ≤ 4 g/dL, pre-albumin < 20 mg/L, transferrin < 200 mg/dL, BMI < 18.5 kg/m^2^, ECM/BCM index > 1.2, PhAα ≤ 4.5°, and weight loss (WL) > 5% in the last 3 months [26].

### 2.3. Intra- and Post-Operative Data Collection

Intra-operative tumor dissemination was documented using the validated intraoperative mapping of ovarian (IMO)-Script at time of cytoreductive-surgery [27,28]. This also included the presence of ascites, peritoneal carcinosis, and residual tumor. No visible residual tumor was defined as complete cytoreduction. Tumor dissemination pattern was determined through the IMO-Script by quadrants and levels. The final histology, grading, and International Federation of Gynaecology and Obstetrics (FIGO) staging were also evaluated [29].

All complications occurring within 30 post-operative days were recorded. All patients were then followed up for a period of three years, following the primary surgery, to determine relapse, response, and survival rates.

The predictive value of malnutrition in patients with EOC was assessed using three main factors: postoperative residual tumor, postoperative complications and response to platinum-based chemotherapy [27]. Moreover, we evaluated predictive and prognostic values of malnutrition via overall survival (OS) and progression-free survival (PFS) analysis.

### 2.4. Statistical Analysis

The results of the descriptive statistical nominal data were indicated as absolute values and percentages. We performed the Kolmogorov–Smirnov and Shapiro–Wilk tests to check the data’s normality and accepted an alpha value of >0.05 as normally distributed. We calculated mean values together with ±standard deviation (SD) for the normally distributed data and median values, together with the 25th and 75th interquartile for the non- normally distributed data. Ninety-five percent confidence intervals (CIs) were presented, where applicable. To analyze the differences between patient groups, we used Pearson’s chi-square test for nominal data. For continuous variables, group distributions were calculated using the Mann and Whitney U-test. ROC curves were used to calculate the diagnostic accuracy of the methods used for nutritional-status assessment, with the NRS-2002 as a reference standard. For all other indicators of nutritional status that were used (NRI, PhA, ECM/BCM index, albumin, prealbumin, transferrin, and 5% weight loss in the last three months) a specific cut-off value was defined, which was subsequently used to classify patients’ nutritional status. A phi-factor 0,8 was used to classify a test as redundant. We performed univariate and multivariate survival analyses using the Kaplan–Meier method (log-rank testing) and Cox regression models, respectively. In this study, we accepted *p*-value < 0.005 as statistically significant. The results of the multivariate logistic regression analyses are given as *p* value, odds ratio (OR), and 95% confidence interval (95% CI). We used the Statistical Package for the Social Sciences (SPSS) software, version 19.0 for Windows (SPSS Inc., Chicago, IL, USA) for the data analysis.

## 3. Results

Our final analysis included a total of 152 patients. Among these patients, 52% (n = 79) had primary, and 48% (n = 73) had recurrent EOC. The median age was 56 (19–84) years. Table 1 represents the patients’ baseline characteristics.

The validity of each nutritional-status indicator was evaluated in comparison with the NRS-2002. All nutritional-status indicators correlated significantly with the NRS-2002 (*p* < 0.001), apart from BMI (*p* = 0.786). The respective sensitivity and specificity values of each method are shown in Table 2. According to the NRS-2002, a total of 18.4% (n = 28) of patients were classified as malnourished. Of those, 18 (64.3%) patients had primary EOC and 10 (35.7%) had recurrent EOC. Consequently, 22.8% of patients with primary EOC and 13.7% of patients with recurrent EOC were classified as malnourished. Moreover, depending on the nutritional-status indicator used, between 2% and 78.1% of patients with EOC were classified as malnourished (Table 2).

### 3.1. Risk Factors for Malnutrition

Patient and tumor characteristics were evaluated according to NRS-2002. Patients > 65 years old, with ascites of >500 mL, or with platinum-resistant EOC showed a statistically significant increased risk of malnutrition when evaluated using NRS-2002 (*p*-values = 0.014, 0.001, and 0.007, respectively) (Table 3). There were no other tumor or patient characteristics that correlated significantly with NRS-2002 ≥ 3 (Table 3).

The tumor characteristics were then correlated with the other nutritional-status indicators and with the NRS-2002, as demonstrated in Table 4.

Non-serous and high grade EOC, bowel infiltration, peritoneal carcinomatosis, and advanced FIGO stage were associated with increased risk of malnutrition. The NRI showed similar correlation with the NRS-2002 but was not compatible with regards to tumor type, bowel involvement, and peritoneal carcinomatosis. Pre-albumin correlated with NRS-2002 with regard to ascites and platinum-sensitivity, but did not show correlation with age. Transferrin correlated with NRS-2002 with regard to age, ascites and response to platinum-based chemotherapy. Moreover, transferrin correlated with bowel-involvement, peritoneal carcinosis and advanced FIGO stage. WL over 5% in the last 3 months correlated with age but not with ascites and platinum-sensitivity. The ECM/BCM index corresponded to the NRS-2002, tumor stage, grading and FIGO-stage, also influencing the ECM/BCM index. The PhAα correlated with age and platinum-sensitivity similarly to the NRS-2002, but not with regard to ascites. Albumin was compatible with the NRS-2002 with regard to ascites, platinum-sensitivity and age. In addition, low albumin levels also correlated with bowel involvement.

On average, all patients had tumor in at least three quadrants. Patients with NRS-2002 ≥ 3 had a significantly higher tumor burden than patients with NRS-2002 < 3 (*p* = 0.044) (Table 5). All nutritional-status indicators correlated significantly with tumor dissemination, except for WL > 5% in 3 months (Table 5).

Moreover, nutritional-status indicators and levels of tumor spread were examined. NRS-2002 showed no significance in tumor dissemination at specific levels in malnourished patients versus non-malnourished. Pre-albumin < 20 mg/L was the only indicator that correlated significantly with a more frequent tumor dissemination on all three levels (level 1, 2, and 3: *p*-values 0.001, 0.006, and 0.018, respectively). Transferrin < 200 mg/dL correlated with more frequent tumor dissemination at level 2 and 3 (*p*-values: 0.001 and <0.001, respectively). Furthermore, albumin ≤ 4.0 g/dL, NRI < 100, and ECM/BCM > 1.2 were correlated significantly with malnutrition at level 3 (*p*-values: 0.008, 0.004, and 0.045, respectively).

### 3.2. Predictive Value of Malnutrition

#### 3.2.1. Cytoreduction

Complete cytoreduction was achieved in 61.8% (n = 94) of cases. Only 32.1% (n = 9) of patients classified as malnourished had complete cytoreduction. The majority of patients (69.1%, n = 85) who had complete cytoreduction had an NRS-2002 < 3. An NRS-2002 < 3 was an independent predictive factor for complete tumor resectability (*p* = 0.009). Patients with NRS-2002 ≥ 3 had a 4.6-fold higher risk of postoperative residual tumor compared to patients with NRS-2002 < 3 (*p* = 0.009, OR = 0.22, 95% CI = 0.07–0.69), based on multivariate logistic regression analysis. The PhAα, NRS-2002, ECM/BCM index, NRI, albumin, and prealbumin showed a significant correlation with postoperative residual tumor. This correlation was mostly significant for the PhAα. When compared to PhAα > 4.5, a PhAα ≤ 4.5 indicated a 5.4-fold higher risk of postoperative residual tumor. WL and transferrin did not correlate significantly with postoperative residual tumor.

#### 3.2.2. Blood Transfusion

More than three-quarters of the patients (75.7%) received at least one unit of packed red blood cells (RBC) during the admission, with a median of two units RBC per patient. The majority of the patients (80.3%) received a blood product (RBCs, fresh frozen plasma, or platelets) with a median of eight units per patient. Malnourished patients received on average more RBC transfusions than non-malnourished patients (*p* = 0.019). On average, malnourished patients received five units RBC (range 0–18) whilst non-malnourished patients received only two units (0–19) (*p* = 0.002).

#### 3.2.3. Postoperative Complications

More than forty-two percent of the patients had at least one postoperative complication, including fistula, ileus, bowel perforation, anastomotic leaks, wound dehiscence, pneumothorax, embolism, infections, sepsis, organ failure, cardiac problems, and postoperative ascites or pleural effusions. The most frequent complication was infection, which occurred in 7.9% patients (n = 12). Malnourished patients were significantly more likely to have a postoperative complication compared to non-malnourished patients (*p* = 0.010). However, in the multivariate regression analysis, the outcome for NRS-2002 was not statistically significant. Instead, only two nutritional-status indicators were independent predictors for postoperative complications: transferrin < 225 mg/dL (*p* =0.003, OR = 3.49, 95% CI = 1.53–7.96) and PhAα ≤ 4.5 (*p* = 0.034, OR = 2.98, 95% CI = 1.09–8.14). Patients with transferrin < 225 mg/dL had a 3.5-fold higher risk of postoperative complications. Patients with PhAα ≤ 4.5 had a 3-fold higher risk of postoperative complications. The other indicators did not show any statistically significant correlation with frequency of postoperative complications.

#### 3.2.4. Mortality and Hospital Stay

In total, three patients died within 30 days after cytoreductive surgery, where the mortality rate was 2.0%. All of them had an NRS-2002 score ≥ 3 and were also classified as malnourished according to pre-albumin, transferrin, the ECM/BCM index and PhAα.

The median hospital stay was 15 days following surgery (2–68 days). There was no significant difference in hospital stay between patients with NRS-2002 ≥ 3 compared to NRS-2002 < 3. However, the ECM/BCM index, NRI, albumin, and pre-albumin correlated significantly with increased duration of hospital stay (*p*-values = 0.007, 0.005, 0.004, and 0.007, respectively). In fact, patients classified as malnourished according to those indicators stayed on average 2–4 days longer compared to non-malnourished patients.

#### 3.2.5. Platinum Response

According to NRS-2002, 18 patients (22.8%) were classified as malnourished. Four of them (22.2%) developed platinum resistance on follow-up. From the non-malnourished patients, only seven patients (11.5%) were platinum-refractory at follow-up. In the univariate analysis, NRS-2002 showed no significant correlation with a worse response to chemotherapy. WL of 5% over the last 3 months was the only indicator that did show a significant correlation with platinum-based chemotherapy response. This was also confirmed in the multivariate regression analysis (*p* = 0.041, OR = 6.99, 95% CI = 1.08–45.45). Consequently, all other indicators were not independent prognostic factors for response to platinum-based chemotherapy.

### 3.3. Prognostic Value of Malnutrition

The follow-up period consisted of 37 months (0–59 months). Estimated median OS was 41 months (95% CI = 33–48 months). After three years, 37 patients (24.3%) had no recurrence. The estimated PFS was 15 months (95% CI= 12–18 months).

The patients with NRS-2002 ≥3 had a median OS of seven months (95% CI = 0–24 months), as compared to the patients with NRS-2002 < 3, where median OS was 46 months (*p* = 0.001). However, in the multivariate regression analysis, the NRS-2002 score was not a significant independent prognostic factor (*p* = 0.051). PhAα and ECM/BCM index correlated with shorter OS, while PhAα ≤ 4.5 was the strongest predictor (Figure 1A,B). No other indicators showed significant correlation with OS.

The malnourished patients, according to NRS-2002, had a PFS of 7 months (95% CI = 2–12 months), compared to non-malnourished patients with a PFS of 16 months (95% CI = 12–20 months). This was statistically significant (*p* = 0.006) (Figure 2). Although PhAα, ECM/BCM index, and transferrin correlated with PFS, they were not statistically significant.

In patients without ascites, we detected a statistically significant difference in OS between malnourished and non-malnourished patients (*p* = 0.001). However, this difference was not statistically significant in patients with ascites < 500 mL and ascites > 500 mL. Hence, as ascite volume increases, the difference in OS between malnourished and non-malnourished patients decreases.

## 4. Discussion

In this prospective observational study, we included 152 patients with EOC and tubal or peritoneal cancer. We classified 28 patients (18.4%) as malnourished according to the NRS-2002. When evaluated using NRS-2002, patients > 65 years-old, with ascites of >500 mL, or with platinum-resistant EOC had a statistically significant increased risk of malnutrition. The majority of nutritional-status indicators correlated significantly with tumor dissemination. The only indicator that correlated significantly with a more frequent tumor dissemination on all three levels was pre-albumin < 20 mg/L. Compared to patients with NRS-2002 < 3 or PhAα > 4.5, patients with NRS-2002 ≥ 3 or PhAα > 4.5 had around a 5-fold higher risk of postoperative residual tumor. Transferrin < 225 mg/dL and PhAα ≤ 4.5 were found as independent predictors for postoperative complications. Only WL of 5% over the last 3 months showed a significant correlation with platinum-based chemotherapy response. While both PhAα and ECM/BCM index correlated with shorter OS, PhAα ≤ 4.5 was the strongest predictor. Based on NRS-2002, the malnourished patients had shorter PFS compared to non-malnourished patients. Among patients without ascites, the malnourished group had longer OS than non-malnourished patients.

Many studies evaluating malnutrition in patients with gynecological cancers often analyze EOC patients as a separate “high risk” subgroup. To evaluate and identify malnutrition, several diagnostic methods are listed in the current literature, including dual-energy X-ray absorptiometry (DEXA) scan and densitometry [30]. These methods are precise, but time-consuming, invasive, and expensive, and as such are not widely used in clinical practice. Various nutritional assessment tools are used, but no accepted gold standard exists. Nonetheless, evaluating and identifying malnutrition within clinical practice remains difficult, due to a lack of diagnostic criteria and consistent documentation. Systematic reviews evaluating frequently used nutrition screening tools warranted a need for a gold standard for use in the elderly [9,17,18,22,23].

The nutritional parameters used in our study were cost-effective, non-invasive, and simple to use. Other nutritional parameters such as the patient-generated subjective global assessment (PG-SGA) and the subjective global assessment (SGA) have been used in studies, to classify between 0 and 81.4% of EOC patients as malnourished [14,15,16,17,18,19,31]. Moreover, we did not use SGA and PG-SGA in our study. This could indicate that our patient cohort was less malnourished compared to the literature, or more likely that the use of the NRS-2002 was more specific to our cohort.

Similar to our findings, increasing age has been previously identified as a risk factor for malnutrition [30]. However, the correlations shown in our study between malnutrition and ascites volume, and malnutrition and platinum-resistance in patients with ovarian cancer have been infrequently reported [32]. Possible explanations could include the fact that ascites predisposes patients to higher protein loss, change in bowel-motility and malabsorption, and an increase in resting energy expenditure (REE) [32].

It is unclear why platinum-resistant patients are at higher risk of malnutrition than platinum-sensitive patients. Perhaps the platinum-resistant tumor biology, due to its worse prognosis and more aggressive natural course, causes metabolic changes leading to malnutrition [31,33,34] (Eisenkop, 2001 #32).

Lieffers et al. reported a direct correlation between tumor mass and REE in patients with advanced colorectal cancer [35]. Similarly, Cao et al. described a higher REE in oncological patients with malnutrition [30,36]. It appears that there is a correlation between tumor mass, tumor dissemination, and malnutrition. However, this correlation has, until now, not been described in EOC patients. In our study, tumor dissemination was documented according to the validated IMO Script [27,28]. We showed that patients who had widely disseminated tumor had a higher risk of malnutrition compared to patients with localized tumor spread. However, the IMO Script does not differentiate between tumor dissemination and tumor burden as a cause for malnutrition.

It is well-established that complete macroscopic cytoreduction is the single most important prognostic factor in EOC patients [37,38,39]. Few studies have been published that examine the correlation between malnutrition and complete cytoreduction. In a prospective observational study of seventy patients with EOC, a low phase angle (reflecting deranged body composition) was significantly associated with incomplete cytoreduction and postoperative complications(38). In addition, in a retrospective study of patients with EOC and who were over 75 years old, an albumin level ≤ 3.7 g/dL was associated with a 2.4-times higher risk of residual tumor at cytoreductive surgery [40]. Our study also shows a similar significant correlation. In our cohort, malnourished patients as classified by NRS-2002 had a higher risk of residual tumor at cytoreductive surgery (OR = 4.6, 95% CI = 1.5–14.5). This is perhaps not surprising, as malnutrition is associated with other factors known to be linked to complete cytoreduction, such as tumor dissemination (especially level 3) platinum-resistance, and ascites [40,41,42].

In our cohort, malnourished EOC patients received on average twice the amount of blood products as non-malnourished patients. Intraoperative blood transfusion increases the likelihood of postoperative complications, including mortality and increased hospital-stay [43]. This could be due to immunosuppression caused after allogenic blood transfusion increasing the risk for infection, as described in patients with colorectal cancer [43]. Another study reported that the surgical patients receiving blood transfusions were at higher risk of anastomotic leaks and postoperative sepsis and have a shorter OS [44]. Consequently, blood transfusions in surgical patients have a negative predictive and prognostic influence.

In addition, it is well reported that postoperative complications are more frequent in malnourished patients [34,41,42].

In our study, malnourished patients had a higher rate of postoperative complications. All EOC patients that died within 30 postoperative days had been classified as malnourished according to the NRS-2002 and other parameters. Moreover, malnutrition significantly correlated with increased hospital stay as classified by NRI, albumin, prealbumin, and the ECM/BCM index. Similar results have been reported in other studies [12,34,41].

Malnourished patients have higher complication rates under chemotherapy treatment and worse response rates compared to non-malnourished patients [45]. In our study, patients with NRS-2002 ≥ 3 were more likely to develop platinum resistance, although, in multivariate regression analysis, this was not statistically significant; interestingly, WL > 5% in the last 3 months significantly correlated with platinum resistance (*p* = 0.041, OR = 6.99, 95% CI = 1.08–45.45). It can be described as a parameter to predict the response to chemotherapy.

Our analysis did not show an independent correlation between preoperative malnutrition and PFS. However, we report that malnutrition as assessed by PhAα ≤ 4.5 is an independent prognostic factor for OS in EOC patients. This correlates with previous reported outcomes of prospective studies [45,46]. In our cohort, EOC patients with ascites had a higher risk of malnutrition. Based on our analysis, ascites was found to increase the risk of malnutrition and indirectly influence the prognosis. Malnutrition seems to be a stronger prognostic factor than ascites for OS in EOC patients. This requires further prospective analyses and was outside the scope of our study.

Our study presents both limitations and strengths. While being a single-center study, its prospective design stands out as a key strength. Additionally, the utilization of multiple validated malnutrition screening tools adds robustness to our findings. However, the relatively small sample size restricts the generalizability of results to larger cohorts. Notably, we did not explore the interplay of various variables, such as the correlation between BMI, malnutrition, and survival outcomes in this study. Nevertheless, our research demonstrates the utility of basic, validated malnutrition assessment tools as predictive and prognostic indicators for patients with EOC. To enhance our understanding, larger prospective studies are warranted to investigate optimal strategies for assessing and preventing malnutrition, especially within the evolving frameworks of prehabilitation and enhanced recovery after surgery (ERAS) protocols in perioperative care for patients with EOC.

## 5. Conclusions

According to the results of our prospective study, malnutrition is an independent predictor of incomplete cytoreduction in patients with ovarian and peritoneal cancer. Moreover, as defined by PhAα, it is also an independent prognostic factor for poor overall survival. The preoperative nutritional assessment is an effective tool in the identification of high-risk groups within patients with ovarian or peritoneal cancer, characterized by poor clinical outcome.

In patients with EOC, malnutrition is a common and serious problem that is often underestimated and misdiagnosed. Our study shows that nutritional status parameters can be used in daily clinical practice to objectively assess malnutrition. Malnutrition plays a significant predictive and prognostic role in the perioperative care of patients. Hence, nutritional status assessment should be standardized and included in preoperative screening to provide nutritional support, improve prognosis, and reduce the consequences of cancer-associated nutritional decline.

## Figures and Tables

**Figure 1 cancers-16-00622-f001:**
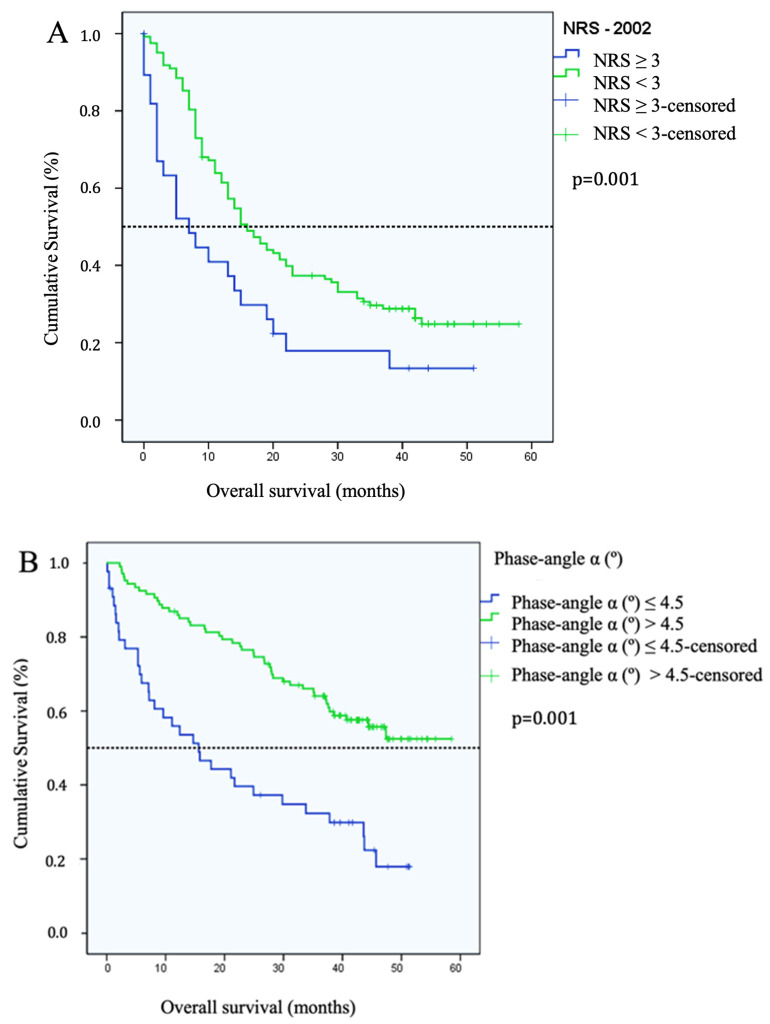
(**A**) Median overall survival of patients with Nutritional Risk Screening (NRS)-2002 ≥ 3 vs. NRS-2002 < 3 (*p* = 0.001) where Kaplan–Meier analysis curve was created according to NRS 2002. (**B**) Median overall survival of patients with phase angle α ≤ 4.5° vs. phase angle α > 4.5° (*p* = 0.001) where Kaplan–Meier analysis curve was created according to phase angle α.

**Figure 2 cancers-16-00622-f002:**
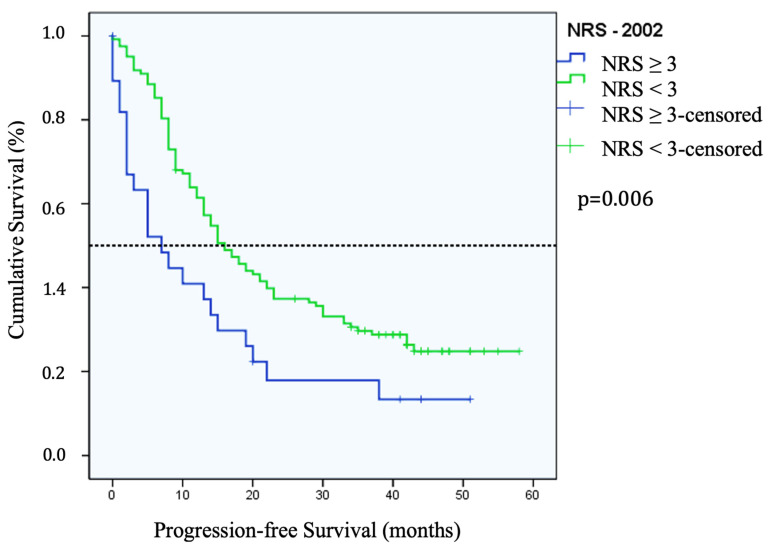
Median progression-free survival in patients with Nutritional Risk Screening (NRS)-2002 ≥ 3 vs. NRS-2002 < 3 (*p* = 0.006) where Kaplan–Meier analysis curve was created according to NRS-2002.

**Table 1 cancers-16-00622-t001:** Patient Baseline Characteristics (n = 152).

Characteristic	Number (%)
Age (years)	56 (19–84) *
Weight (kg)	65 (45–141) *
BMI (kg/m^2^) underweight (<18.5) normal weight (18.5–24.9) overweight (≥25.0) obesity (≥30.0)	24.4 (17.8–48.8) * 17.97 (0.15) (n = 3, 1.97) 19.03 (0.25) (n = 82, 53.95) 25.10 (0.10) (n = 44, 28.95) 30.00 (30.4–35.2) (n = 23, 15.13)
Primary OC	79 (52.0)
FIGO Staging (Primary OC only)	
I	8 (10.3)
II	8 (10.3)
III	39 (50)
IV	22 (28.2)
Unknown	2 (2.5)
Recurrent OC	73 (48.0)
Platin Response (Recurrent OC only)	
Platin sensitive	48 (65.8)
Platin resistant	25 (34.2)
Grading	
I	4 (2.6)
II	40 (26.3)
III	82 (53.9)
Unknown	26 (17.1)
Histology	
Serous	119 (78.3)
Endometrioid	7 (4.6)
Mucinous	6 (3.9)
Clear cell	7 (4.6)
Other	3 (2.0)
Unknown	10 (6.6)
Ascites	
≥500 mL	26 (17.1)
<500 mL	49 (32.2)
No ascites	75 (49.3)
Unknown	2 (1.3)
Tumor Spread	
Small bowel involvement	56 (36.8)
Large bowel involvement	83 (54.6)
Peritoneal carcinomatosis	120 (78.9)
Residual Tumor	
None	94 (61.8)
≤1 cm	30 (19.8)
>1 cm	28 (17.7)

* Median (interquartile range); BMI: body mass index, FIGO: International Federation of Gynaecology and Obstetrics, OC: Ovarian Cancer.

**Table 2 cancers-16-00622-t002:** Prevalence of malnutrition according to various nutritional status indicators and their respective sensitivity and specificity when correlated with NRS-2002.

Nutritional Status Indicator *	Cut-Off Value for Malnutrition	Number (%)	Area under the ROC Curve	Sensitivity (%)	Specificity (%)	CI (95%)
NRS-2002	≥3	28 (18.4)	NA	NA	NA	NA
Prealbumin (mg/L)	<20	51 (37.2)	0.807	77.8	72.7	0.708–0.906
NRI	<100	47 (31.8)	0.801	67.9	76.7	0.707–0.896
Weight Loss in last 3 months (%)	>5	29 (19.1)	0.780	64.3	91.1	0.665–0.895
Transferrin (mg/dL)	<200	41 (28.1)	0.785	65.4	80	0.680–0.890
ECM/BCM Ratio	>1.2	58 (38.4)	0.762	77.8	70.2	0.653–0.871
Phase angle α (°)	≤4.5	44 (29.1)	0.760	66.7	79	0.651–0.869
Albumin (g/dL)	≤4.0	53 (35.3)	0.769	75	73.8	0.665–0.872

BCM: Body Cell Mass, BMI: Body Mass Index, CI: Confidence Interval, ECM: Extra Cellular Mass, NA: not applicable, NRI: Nutritional Risk Index, NRS-2002: Nutritional Risk Screening-2002; * BMI did not correlate with NRS-2002 and was therefore excluded from further evaluations.

**Table 3 cancers-16-00622-t003:** Prevalence of malnutrition as evaluated by the Nutritional Risk Screening (NRS)-2002, according to patient and tumor characteristics.

Characteristic	Label	Total (n = 152) (%)	Patients with NRS ≥ 3 (n = 28) (%)	*p*-Value
Age	>65 years	42 (27.6)	13 (46.4)	*p* = 0.014
	≤65 years	110 (72.3)	15 (53.6)	
Diagnosis	Primary	79 (51.9)	18 (64.3)	NS
	Recurrent	73 (48.0)	10 (35.7)	
Ascites	>500 ml	28 (18.4)	11 (39.3)	*p* = 0.001
	<500 ml	124 (81.6)	17 (60.7)	
Histology	Serous	123 (80.9)	22 (78.6)	NS
	Non-serous	29 (19.1)	6 (21.4)	
Grading	I + II	50 (32.9)	9 (32.1)	NS
	III	87 (57.3)	18 (64.3)	
Bowel involvement	Yes	93 (61.2)	17 (60.7)	NS
	No	59 (38.8)	11 (39.3)	
Peritoneal carcinomatosis	Yes	120 (78.9)	24 (85.7)	NS
	No	30 (19.7)	4 (14.2)	
FIGO Stage	I + II	16 (10.5)	3 (10.7)	NS
	III + IV	63 (41.4)	15 (53.6)	
Platinum sensitivity	Platinum sensitive	49 (32.2)	3 (10.7)	*p* = 0.007
	Platinum resistant	24 (15.8)	7 (25.0)	

FIGO: International Federation of Gynaecology and Obstetrics, NS = not significant.

**Table 4 cancers-16-00622-t004:** Correlation of Nutritional Status Indicators and tumor characteristics, where plus sign (+) indicates positive correlation.

	Nutritional Status Indicators						
	Prealbumin (<20 mg/L)	NRI (<100)	Weight Loss in Last 3 Months (>5%)	Transferrin (<200 mg/dL)	ECM/BCM (>1.2)	Phase-Angle α (≤4.5°)	Albumin (≤4.0 g/dL)
Indicator Tumor Characteristics							
Age		+	+	+	+	+	+
Ascites	+	+		+	+		+
Platinum Sensitivity	+	+		+	+	+	+
Primary/Recurrent		+	+		+		
Histology						+	
Grading					+		
Bowel Involvement	+	+		+			+
Peritoneal carcinomatosis	+	+		+			
FIGO Stage	+			+	+		

BCM: Body Cell Mass, ECM: Extra Cellular Mass, FIGO: International Federation of Gynaecology and Obstetrics.

**Table 5 cancers-16-00622-t005:** Malnutrition and tumor spread according to Nutritional Status Indicators.

Nutritional Status Indicators	Number of Fields with Tumor Load—IMO Script (Median)		*p*-Value
	Malnourished	Non-Malnourished	
NRS–2002 (≥3)	5	3	0.044
NRI (<100)	6	3	<0.001
Prealbumin (<20 mg/L)	6	3	<0.001
Transferrin (<200 mg/dL)	6	3	<0.001
Albumin (≤4.0 g/dL)	5	3	0.001
ECM/BCM (>1.2)	4	3	0.024
Phase angle α (≤4.5°)	4	3	0.041
Weight loss in last 3 months (>5%)	4	3	NS

BCM: Body Cell Mass, BMI: Body Mass Index, ECM: Extra Cellular Mass, IMO: Intraoperative Mapping of Ovarian cancer, NRI: Nutritional Risk Index, NRS-2002: Nutritional Risk Screening 2002, NS = not significant.

## Data Availability

The data presented in this study are available on request from the corresponding author.

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
