# Peer review of "Pre-Operative Malnutrition in Patients with Ovarian Cancer: What Are the Clinical Implications? Results of a Prospective Study"

_cancers, 2024, doi:10.3390/cancers16030622_

Round 1

Reviewer 1 Report

Comments and Suggestions for Authors

This is an interesting research article with quite novelty. However, some points should be revised and improved.

- In summary and abstract section, there are several English misspelings and typos errors.  

- The introduction section is quite small. The authors should report some previous studies on their topic, emphasizing the litarature gap for which they performed the present study to cover this literature gap.

- In the Assessment of Nutritional Status section, the authors should add some relevant references at the end of the first sentence (line 74).

Accordingly, the sentence in lines 131-132 need a relevant reference.

- In the statistical analysis section, the authors should report what statistical test was used for evaluating the difference, e.g. Chi-square test, Mann-Withney test, e.t.c.

- In Table 1, BMI could be categorized according to the recommended criteria into 4 groups: underweight, normal weight, overweight and obesity.

- May the intoduction oft he categorized BMI variable in Tables 3 and 5 offer some useful information?

- In Figures 1 and 2, p-values should be included into the figure.

- At the end of the discussion section, the authors should add at least one paragraph with the strengths and the limitations of their study.

- Moderate English language editing is required.

Comments on the Quality of English Language

Moderate editing of English language required

Reviewer 2 Report

Comments and Suggestions for Authors

Dear Authors,

my comments:

1. what type of study did you perform? Prospective study is not enough.

2. In my opinion data  and references are too old.

3. Technical problems with tables 4 and 5.

4. Inclusion and exclusion criteria should be described more precisely.

Round 2

Reviewer 1 Report

Comments and Suggestions for Authors

The authos have significantly tried to improve their manuscript. However, several points should be addressed before a final desicion.

- Line 81: ".... were found to be 19 times more likely to ...". Is this correct (19 times) or do you mean 1.9 times?

- Line 162-163: Please add a relevant reference for PhAα cut-off.

- Lines 175-176: Please add a relevant reference for FIGO staging.

- In lines 192-193 the authors report that: "For continuous variables, group distributions were calculated using the Mann and Whitney U-test." However, in this case all variables should be non-normally distibuted. The authors should firstly report which statistical test used for assesing the normality of variables. Secondly, the authors should use and report that for the normally distributed contunious variables student t-test was applied. Moreover, the authors should use and report that the non-normally distributed contunious variables were expressed as median and interquartile (IQR) and that the non-parametricn Mann-Whitney U-test was applied.

- Accordingly, in the Table 1, median values and IQR were used. Does this mean that the variables were non-normally distributed. This is confusing concerning the previous statements of authors in the statistical analysis section.

- In Table 1, ordinal or categorical variables were also included. This should be reported in the statistical analysis section, mentioning also how these variables are expressed.

- In Table 3, all the variables are categorical. What statistical test was used? This should be reported in the statistical analysis section.

- Again, in Table 5, what statistical test was used?

- Overall, concerning the statistical analysis of the study data, a more careful statistical analysis should be performed by an expert statistician. Otherwise, the results of the study will present serious scientific flaws.

- In line 163, Please revise the sentence "...using the Kaplan–Meier method (log-rank testing) and Cox regression models." as "...using the Kaplan–Meier method (log-rank testing) and Cox regression models, respectively."

- The "Predictive Value of Malnutrition" is quite compex and confusing for the reader. The author should add subheadings in this section.

- In the "Prognostic Value of Malnutrition", the authors firstly reported that "The follow-up period consisted of 37 months (0-59 months)". There is a discrepancy here.

- Moreover,in the same section the authors reported that "The patients with NRS-2002 ≥3 had a median OS of seven months (95% CI=0-24 351 months)". In this case IQR should be reported.

- The legends of Figures 1 and 2 should report that "Kaplan-Meier analysis curves according to......".

- Lines 300-301, The authors report that "An NRS-2002 <3 was an independent predictive factor for complete tumor resectability (p=0.009)." This was derived by multivariate analysis based on Cox-regression analysis? If yes, this should be reported to this point and in the relevant following results.

- In the Discussion section, the first paragraph is a quite long extent repetition of the results of the study. The authors correclty begin the discusion section with their results but this should be more concise reporting only the most significant results.

- Line 413, Please add relevant references. These may be previously reported but they should also reported to this point.

- To estimate BMI, body weight and height data are required. These data were measured or self-reported. This should be noted in lines 135-139. Moreover, if body weight and height data were self-reported, then this should be reported in the limitations study. Overall, any self-reported data may have recall bias, which should be mentioned as a limitation of the study.

- BMI is a good indicator for a general overview of fat mass but does not reflect the fat distribution. Waist circumference and waist to hip ratio were related with abdominal obesity which is higher associated with the risk of several diseases. This should also be reported.

- Moderate English language editing is still required throughout the manuscript.

Comments on the Quality of English Language

Moderate English language editing is still required throughout the manuscript. There are still several English misspelings and grammar/syntax errors which should be revised by a native English speaker/writer.

Reviewer 2 Report

Comments and Suggestions for Authors

I accept this second version and thank you for you reply.

Author Response

Thank you for your revisions and feedback to ímprove the manuscript. 

Round 3

Reviewer 1 Report

Comments and Suggestions for Authors

The authors have now significantly improved their manuscript. 

I would like to thank the authors for your trust to mu suggestions.